# Diagnoses and critical care outcomes in a rural Tanzanian high dependency unit: A prospective cohort study

Andrew Katende[1,2ʘ], Julie Rossier[3,4ʘ], Chipegwa Mlula[1,2], Christamonica Chitimbwa[1,2], Martin E. Mtula[1,2], Elibariki S. Mafole[1,2], Lulu Wilson[1,2], Samuel S. Mutasingwa[1,2], Evance Mahundi[1,2], Mohamed K. Kipalaunga[1,2], Victor Myovela[1,2], Caspar Mbawala[1,2], Fanuel Faustine[1,2], Geofrey Mbunda[1,2], Winfrid Gingo[5], Faraja Kitila[6], Ipyana Mwasongwe[6], Claudia Bucher[4,6,7], Daniel H. Paris[4,7], Thomas Zoller[8], James Okuma[4,7], Maja Weisser[1,4,7,9], Martin Rohacek[1,2,4,6,7]*

**1** Department of Interventions and Clinical trials, Ifakara Health Institute, Ifakara, United Republic of Tanzania, **2** Heart and Lung Clinic and High Dependency Unit, St. Francis Regional Referral Hospital, Ifakara, United Republic of Tanzania, **3** Intensive Care Unit, University Hospital Basel, Basel, Switzerland, **4** University of Basel, Basel, Switzerland, **5** Department of Surgery, St. Francis Regional Referral Hospital, Ifakara, United Republic of Tanzania, **6** Emergency Department, St. Francis Regional Referral Hospital, Ifakara, United Republic of Tanzania, **7** Department of Medicine, Swiss Tropical and Public Health Institute, Allschwil, Switzerland, **8** Department of Infectious Diseases and Respiratory Medicine, Charité – Universitätsmedizin Berlin, corporate member of Freie Universität Berlin and Humboldt-Universität zu Berlin, Berlin, Germany, **9** Division of Infectious Diseases, University Hospital Basel, Basel, Switzerland

ʘ Authors contributed equally to the manuscript
* martin.rohacek@swisstph.ch; mrohacek@ihi.or.tz

## Abstract

### Background

Data on rural sub-Saharan African high-dependency units (HDU) are lacking. We describe patient's characteristics, diagnoses, and outcomes of patients admitted to a Tanzanian HDU, and identified factors associated with in-hospital mortality.

### Methods

This prospective single-center cohort study was conducted in the HDU of a Tanzanian rural referral hospital. All patients admitted to the HDU were eligible. Descriptive analyses, and univariate and multivariate modeling to identify predictors of in-hospital mortality were done. Kaplan-Meier survival curves were employed to estimate mortality rates over time. The area under the receiver operating characteristic curve was used to assess the predictive accuracy of early warning scores.

### Results

From April 4th 2023 to March 29th 2024, 491 patients were included and followed-up until hospital discharge. Median age was 46 years (IQR 29−65); 259 (53%) were females. Most common diagnoses were sepsis (N=96, 20%), arterial hypertension

**Data availability statement:** All relevant data are within the paper and its Supporting information. The datasets used and/or analyzed during the current study are available with restricted access from Zenodo at https://zenodo.org/records/15177213.

**Funding:** Martin Rohacek received a grant from the Else Kröner-Fresenius-Stiftung, Bad Homburg vor der Höhe, Germany, for the implementation of the high dependency unit at the St Francis Regional Referral Hospital, Ifakara. Julie Rossier received financial support from the Freiwillige Akademische Gesellschaft Basel, Basel, Switzerland, to finance her travel- and living expenses during her stay in Ifakara. The funders had no role in study design, data collection and analysis, decision to publish, or preparation of the manuscript.

**Competing interests:** The authors have declared that no competing interests exist.

(N = 91, 19%), diabetes mellitus (N = 84, 17%), acute kidney injury (N = 66, 13%), decompensated heart failure (N = 64, 13%), aspiration pneumonia (N = 60, 12%), and stroke (N = 59, 12%). Mortality during HDU- and hospital stay was 30%(N = 146) and 37%(N = 182), respectively. 54% of patients with sepsis, 51% with stroke, 65% with aspiration pneumonia and 27% with heart failure died in the HDU. Predictors of in-hospital mortality were age ≥ 45 years versus 18−44 (adjusted Hazard Ratio (aHR) 1.56, 95% CI 1.07–2.28, p = 0.03), blood pressure <90mmHg (aHR 2.33, 95%CI 1.48–3.81, p < 0.001), Glasgow Coma Scale score ≤8 versus 14−15 (aHR 2.13, 95%CI 1.24–3.64, p = 0.02) and oxygen saturation at room air < 90% (aHR 1.62, 95%CI 1.04–2.51, p = 0.03). The area under the curve predicting in-hospital mortality was 0.69 (95%CI 0.65–0.73) for the NEWS- and UVA scores, 0.66 (95%CI 0.62–0.70) for the MEWS-, and 0.65 (95%CI 0.61–0.69) for the qSOFA score.

## Conclusion

Sepsis and non-communicable diseases were the most common diagnoses. Scores predicted in-hospital mortality with a moderate accuracy.

## Introduction

Critical care medicine is important to manage seriously ill patients suffering from sepsis, pneumonia, and from non-communicable diseases (NCDs) such as heart failure and stroke [1,2]. Globally, NCDs killed at least 43 million people in 2021, and 73% of these deaths occurred in low- and middle income countries [3]. Critical care services in sub-Saharan Africa increased since the COVID-19 pandemic, but remain limited compared to high-income countries [4–6]. Intensive care units (ICUs) are few [7,8] and are restricted to major urban centers [4]. In remote areas in sub-Saharan Africa, people often present in advanced stages of diseases, and in critical conditions [9–12]. This is attributable to delayed diagnosis, geographical barriers that affect timely health seeking behaviors, and lack of live-saving therapies [6,13]. Postoperative care remain the leading cause for admissions in many ICUs in low- and middle income countries, followed by traumatic brain injuries and medical conditions [14–16]. The outcomes of patients admitted to ICUs varies by region, with mortality rates up to 52% in urban hospitals [15–17]. Replicating standard ICUs in urban centers to rural settings might be challenging due to high costs, inadequate human resources, and lack of expertise [18,19]. Critical care units, tailored to existing health systems in rural areas are warranted. Without these units, critically ill patients admitted to hospital in rural areas are likely to receive inadequate care in general wards [20,21]. High-dependency units (HDU) provide a higher level of care than general wards, but do not reach the level of care provided in ICUs. An HDU can be defined as unit that serves for critically ill patients, but does not provide invasive ventilation [22]. Starting with a HDU as a first step to integrate critical care services into existing hospitals is promising [2]. In sub-Saharan Africa, many hospitals have not implemented these units yet. Data about patient outcomes of ICUs from urban centers are limited, and

data on HDUs situated in rural sub-Saharan Africa are lacking. The objective of this study was to describe characteristics, diagnoses, and outcomes of critically ill patients admitted to a recently implemented HDU of a referral hospital in rural Tanzania, and to identify predictors of in-hospital mortality. This is the first study on outcomes of patients admitted to a HDU in rural sub-Saharan Africa.

## Materials and methods

### Study design and setting

This prospective observational single center cohort study including 491 patients was conducted at the HDU of the St. Francis Regional Referral Hospital (SFRRH), Ifakara, Tanzania.

The SFRRH is a 370-bed regional referral hospital for a rural population of about one million people living in the Kilombero, Malinyi and Ulanga districts in rural Tanzania. It has an emergency department (ED) attending about 90,000 patients per year [23]; 10% of these patients arrive in critical conditions [24].

### Implementation of the HDU

In 2021, at the peak of the second COVID-19 wave, an oxygen plant was installed at SFRRH allowing continuous delivery of oxygen to the ED, surgical theaters, neonatal wards, and a dedicated ward was established to operate as the HDU. Key elements of the HDU implementation starting in 2022 were provision of continuous training for nurses and medical doctors [25], installation of locally adapted and reliable medical equipment [25,26] implementation of standardized monitoring to ensure improved patient outcomes [7,27], and the establishment of a multidisciplinary team [28].

### Training

Daily bedside teaching, seminars, and exchange programs were implemented to strengthen the knowledge in critical medicine of both the hospital staff and HDU team members, utilizing the expertise of both local and visiting specialists.

Two-week training courses in critical care are held every year since 2022, facilitated by experts from the ICU of the University Hospital Basel, Switzerland, and by experts from the ICU of the Jakaya Kikwete Cardiac Institute, Dar es Salaam, and of the Benjamin Mkapa Hospital, Dodoma, United Republic of Tanzania. Theoretical instruction was combined with practical trainings. Specialist ICU nurses spent one week in the HDU to provide training in critical nursing during everyday practice. To allow staff to have an external experience and knowledge transfer, 4 members of the HDU were sent to the ICU of the Muhimbili National Hospital for a 4-week attachment. Regular Medical Council of Tanganyika (MCT)- accredited courses in point-of-care ultrasound (POCUS) including lung ultrasound, echocardiography, and vascular ultrasound were conducted. All medical doctors were trained in POCUS and focused echocardiography.

### Equipment

Medical equipment such as 8 patient monitors, ultrasound- and electrocardiography (ECG) machines, suction devices, and a defibrillator were installed and staff trained in handing before start of the study. The Abbott iSTAT® point-of-care lab system allows rapid electrolyte and blood gas testing, and the stable supply of consumables such as central venous catheters, and chest and pericardial drains was established. A total of 7 ventilators (Dräger Savina 300) used for non-invasive ventilation, and 4 infusion pumps were implemented during the study period.

### Service delivery and organization

The organization at the HDU features a two-shift duty roster for 10 nurses and 4 medical doctors (CM, CC, MEM, ESM,) supervised by two experienced physicians (AK, MR). Clinical rounds occur three times daily, involving nurses, doctors, students, and physicians. An internal HDU staff meeting is held once a month to discuss strategies for enhancing

performance of the unit. Patient management is closely coordinated with other specialized units such as obstetrics-gynecology, surgery, urology, internal medicine, infectious diseases specialists, physiotherapy, orthopedic and dialysis unit. Standard operating procedures for patient admissions and discharge to and from the HDU have been developed and disseminated. HDU doctors evaluate all patients who have been referred for admission to the HDU from other hospital departments, and collaborative decisions on the management are done together with specialists and the HDU team. Patients in need of an emergency computed-tomography (CT) scan are directly transferred to the nearby located Good Samaritan Cancer Hospital. Stable patients are discharged to the general ward with summary reports and a verbal handover. The HDU stores emergency and frequently used drugs, allowing patients to receive the medication needed without immediate payment, facilitating timely intervention, with the cost of the drugs being reimbursed subsequently.

### Participants

All patients admitted to the HDU during the project period were eligible to participate in the study. Exclusion criterion was the refusal to the use of data for research purposes.

### Study procedures and data collection

All patients were assessed upon admission to the HDU, at discharge from the HDU, at discharge from the hospital, and at 30 days post-admission to the HDU using systematic data collection forms. Post-hospital discharge follow-ups were conducted through phone calls. Information on deaths during hospital stay was obtained from the attending physician in the ward or patient files. Data on medical history, physical examination, vital signs, test results (abdominal sonography, POCUS, ECGs, echocardiograms, X-rays, CT scans, laboratory findings) and diagnoses were collected in real-time from patient charts and entered into an electronic locally stored and secured database (Epi Datav4.7.0) by members of the study team. Data were collected by clincians working at the HDU, and responded to queries raised by the data manager and the statistician who cleaned the data. Before the start of the study, all members were trained and instructed how to fill data into the standartized electronic data collection tools. Comprehensive echocardiograms were performed by experienced echocardiographers using a Mindray M7 ultrasound machine equipped with a P 2–5 sector probe. POCUS and focused echocardiograms were done using Mindray M7 or an Echonous Kosmos ultrasound device. ECGs were done using a Schiller Cardiovit MS-2015. Blood gas analyses and serum electrolytes were measured by the iStat point-of-care system (Abbott iSTAT®). Examinations were done upon discretion of the attending physicians. Several early warning scores including Universal Vital Assessment (UVA), National Early Warning Score (NEWS), Modified Early Warning Score (MEWS), and Quick Sequential Organ Failure Assessment (qSOFA) score [29–32] were used to assess patient's severity status.

### Definitions

Sepsis was defined as a life-threatening condition with organ dysfunction due to dysregulated host response to infection, and managed according to current international guidelines [32,33]. Heart failure was defined as presence of symptomatic heart disease confirmed by comprehensive echocardiography, accompanied by dilated jugular veins, lower limb edema, pulmonary edema, or shock, and was managed according to current guidelines [34,35]. Pulmonary edema was defined as presence of bilateral crackles on lung auscultation and presence of bilateral B lines in lung ultrasound [36]. Stroke was defined as acute onset of neurologic symptoms such as hemiplegia, aphasia, gaze deviation, or decreased level of consciousness, and no other explanation for the condition such as electrolyte disorder. If CT of the head showed hypodense lesions as signs of infarction, the etiology was presumed ischemic, if hyperdense lesions indicating hemorrhage were present, stroke was deemed hemorrhagic. If CT was not available, an etiology was not definable. Hypertension was defined as presence of at least 2 documented measurements of elevated blood pressure of ≥140/90 mmHg, or history of

hypertension treatment [37]. Diabetes was defined as random blood glucose of >11 mmol/L, fasting glucose of >7mmol/L, or history of treated diabetes mellitus. Pneumonia was defined as presence of respiratory symptoms and detection of infiltrates, or interstitial changes by ultrasound or chest X-ray [36]. Kidney failure was defined as an estimated glomerular filtration rate (eGFR) of < 60 ml/min.

## Statistical analysis

Descriptive statistics were used to summarize baseline characteristics and diagnoses at admission and discharge. Kaplan-Meier survival curves were employed to estimate and visualize mortality rates over time. Censoring patterns were examined, and the assumption of non-informative censoring was verified. To assess homogeneity between survival groups, log-rank tests were conducted. The association between clinical and demographic factors, including age, sex, occupation, admission source, vital signs, comorbidities, presenting symptoms, and early warning scores (UVA, NEWS, MEWS, qSOFA), and in-hospital mortality was assessed using univariable and multivariable Cox proportional hazards regression. This method accounts for differences in follow-up time and provides adjusted hazard ratios (HRs) to quantify the strength of associations. The proportional hazards assumption was tested using Schoenfeld residuals and log-minus-log plots and was not violated. To compare survival distributions across groups, log-rank tests were performed, and non-informative censoring assumptions were verified. Although we included all eligible patients admitted to the HDU during the study period (i.e., a complete cohort), multivariable analyses were conducted to adjust for potential confounders and to identify independent associations between clinical and demographic variables and in-hospital mortality. These adjusted estimates enhance internal validity and may be generalizable to similar resource-limited clinical settings. To evaluate the predictive performance of clinical scores, we conducted area under the receiver operating characteristic curve (AUROC) analysis, which measures the discriminative ability of each score in predicting in-hospital mortality. All statistical analyses were performed using Stata version 16. A p-value of <0.05 was considered statistically significant.

## Ethical approval and consent to participate

Ethical approval was sought from the Institutional Review Board of the Ifakara Health Institute (IHI/IRB/12–2023), and the National Institute for Medical Research (NIMR/HQ/R.8a/Vol.IX/4247). Patients and their relatives were informed orally that data about their diagnosis and their outcome were used for research purposes. Written informed consent was waived by both ethics' committees.

## Results

### Patient enrollment and follow-up

From April 4th 2023 to March 29th 2024, 516 admissions of 491 patients were recorded and included in the final analysis. All 491 patients were assessed at admission and were followed up for up to 30 days post-discharge from HDU. Forty-five (9.2%) patients were not reachable after discharge from the hospital and were classified as lost to follow up. The median length of stay in the HDU was 2 days (Interquartile range IQR: 1–3), while the overall median hospital stay was 4 days (IQR: 2–8).

### Baseline characteristics on admission and diagnoses at discharge from HDU

Table 1 shows baseline characteristics of all included patients. The median patient age was 46 years (IQR 29–65), 259 (53%) were females, 128 (26%) had a history of hypertension, 85 (17%) had a history of diabetes mellitus, and 35 (7%) were people living with HIV (PLHIV). A total of 280 (57%) patients had been admitted directly from the emergency department, and 120 (24%) were referred after surgery. The main reason for admission to the HDU was respiratory failure in 247 (50%) patients. Other reasons for admission were a trauma in 39 (8%), and postoperative care in 129 (26%) patients.

**Table 1. Patients' characteristics at admission in the high-dependency unit (N = 491).**

| Socio-demographic parameters | N (%) |
|---|---|
| Age (years), median (IQR) | 46 (29-65) |
| Age categories (years) | |
| <18 | 27 (6) |
| 18–44 | 213 (43) |
| ≥45 | 251 (51) |
| Male | 232 (47) |
| Female | 259 (53) |
| Farmer | 385 (78) |
| Non-farmer | 106 (22) |
| **Medical history** | |
| Hypertension | 128 (26) |
| Diabetes mellitus | 85 (17) |
| HIV infection | 35 (7) |
| Heart failure | 28 (6) |
| Stroke | 24 (5) |
| Tuberculosis | 12 (2) |
| Cancer | 9 (2) |
| Asthma | 7 (1) |
| Chronic obstructive pulmonary disease | 6 (1) |
| **Admission** | |
| Emergency department | 280 (57) |
| Theatre | 120 (24) |
| Hospital ward | 91 (19) |
| **Reasons for admission** | |
| Respiratory failure | 247 (50) |
| Neurologic problem | 157 (32) |
| Postoperative care | 129 (26) |
| Hyperglycemia | 43 (9) |
| Trauma | 39 (8) |
| Hypertensive emergency | 33 (7) |
| Septic shock | 13 (3) |
| Diabetic coma | 10 (2) |
| **Surgery (n = 138)** | |
| Laparotomy | 81 (59) |
| Urologic intervention | 23 (17) |
| Chest tube | 5 (4) |
| **Assessment at admission** | |
| Airway problem | 44 (9) |
| Breathing problem | 380 (77) |
| Circulation problem | 339 (69) |
| Disability problem | 309 (63) |
| Exposure problem | 182 (37) |
| BMI (kg/m$^2$), median (IQR) | 23 (21 –25 ) |
| Underweight (BMI < 18.5) | 52 (11) |
| Normal (BMI 18.5 –< 25) | 298 (61) |
| Overweight (BMI 25 –< 30) | 102 (21) |

*(Continued)*

**Table 1.** (Continued)

| Socio-demographic parameters | N (%) |
|---|---|
| Obese (BMI ≥ 30) | 39 (8) |
| GCS score (n = 490) | |
| ≤ 8 | 111 (23) |
| 9-13 | 120 (24) |
| 14-15 | 259 (53) |
| Systolic BP, mmHg, median (IQR)[a] | 125 (109-144) |
| Diastolic BP, mmHg, median (IQR)[a] | 77 (66-92) |
| Hypotension[a] | 41 (8) |
| Hypertension[a] | 191 (39) |
| MAP (mmHg), median (IQR)[a] | 94 (80-109) |
| MAP < 65 mmHg[a] | 38 (8) |
| Heart rate, median (IQR) | 100 (84-119) |
| Heart rate >100/min | 244 (50) |
| Respiratory rate, median (IQR) | 26 (20 –33 ) |
| Respiratory rate >18/min | 409 (83) |
| Oxygen saturation, median (IQR) | |
| On room air | 92 (80-97) |
| On oxygen (n = 234) | 94 (91-97) |
| Oxygen saturation, n (%) | |
| On room air < 90% | 231 (47) |
| On oxygen <90% | 46 (20) |
| Temperature (°C), median (IQR)[b] | 36.2 (36.0-36.8) |
| Temperature > 37.8°C[b] | 39 (8) |
| **Laboratory parameters** | |
| Hemoglobin (g/dl), median (IQR) | 10.2 (7.5-12.2) |
| White blood cells (10³/L), median (IQR) | 9.7 (6.6-16.1) |
| Glucose (mmol/L), median (IQR) | 7.9 (5.9-12.7) |
| Sodium (mmol/L), median (IQR) | 137 (131-142) |
| Potassium (mmol/L), median (IQR) | 4.2 (3.7-5.0) |
| Creatinine (mmol/L), median (IQR) | 95.2 (65.8-200.9) |
| mRDT | 11 (2) |

Data were available from all participants unless marked with a or b.

[a]blood pressure was available from 487 patients.

[b]Body temperature was available from 489 patients. Results are frequency and percent (n (%)) of those with non-missing data, if not indicated otherwise.

IQR-Interquartile range, BMI, Body Mass Index; GCS, Glasgow Coma Scale score; BP, blood pressure; MAP, mean arterial pressure; Hypotension, BP systolic <90mmHg; Hypertension, BP systolic ≥140 mmHg or BP diastolic ≥90 mmHg, HIV-Human immunodeficiency virus, mRDT- Malaria rapid diagnostic test.

At admission a Glasgow Coma Scale (GCS) score of less than 8 was observed in 111 (23%), a mean arterial pressure (MAP) of less than 65 was measured in 38 (8%), and an oxygen saturation of <90% on room air was measured in 231 (47%) patients.

Table 2 shows diagnoses at admission and at discharge from the HDU, and the number of deaths in the HDU. The most common patient diagnoses at discharge from HDU were sepsis (N = 96, 20%), hypertension (N = 91, 19%), diabetes mellitus (N = 84, 17%) acute kidney injury (N = 66, 13%), decompensated heart failure (N = 64, 13%), aspiration pneumonia

**Table 2. Diagnoses at admission, discharge and death among patients in the high- dependency unit.**

| Diagnosis | Admission Total n = 491 N (%) | Discharge Total n = 491 N (%) | Death Total n = 146[a] N (%) |
|---|---|---|---|
| Hypertension | 89 (18) | 91 (19) | 27 (30) |
| Sepsis | 81 (17) | 96 (20) | 52 (54) |
| Diabetes mellitus | 75 (15) | 84 (17) | 13 (15) |
| Heart failure, decompensated | 72 (15) | 64 (13) | 17 (27) |
| Pulmonary edema | 65 (13) | 40 (8) | 15 (38) |
| Stroke | 62 (13) | 59 (12) | 30 (51) |
| Hemorrhage | 15 (24) | 24 (41) | 11 (45) |
| Ischemic | 7 (11) | 10 (17) | 2 (20) |
| Inconclusive | 40 (65) | 25 (42) | 17 (68) |
| Pneumonia, aspiration | 54 (11) | 60 (12) | 39 (65) |
| Pneumonia, CAP | 53 (11) | 28 (6) | 11 (39) |
| Acute kidney injury | 44 (9) | 66 (13) | 33 (50) |
| Trauma brain injury | 27 (6) | 25 (5) | 8 (32) |
| Hypertensive heart disease | 22 (4) | 31 (6) | 10 (32) |
| Seizures | 21 (4) | 14 (3) | 9 (64) |
| Skin ulcers | 16 (3) | 22 (5) | 6 (27) |
| Pulmonary embolism | 15 (3) | 20 (4) | 10 (50) |
| Diabetic ketoacidosis | 15 (3) | 14 (3) | 5 (36) |
| Malaria | 11 (2) | 13 (3) | 7 (54) |
| Peritonitis | 11 (2) | 6 (1) | 5 (83) |
| Exacerbated COPD | 9 (2) | 11 (2) | 4 (36) |
| Delirium | 8 (2) | 10 (2) | 2 (20) |
| Fracture lower limb | 7 (1) | 7 (1) | 4 (57) |
| Intestinal obstruction | 6 (1) | 4 (1) | 1 (25) |
| Trauma abdominal bleeding | 5 (1) | 6 (1) | 2 (33) |
| Intoxication | 5 (1) | 5 (1) | 1 (20) |
| Acute myocardial infarction | 5 (1) | 7 (1) | 6 (86) |
| Asthma | 4 (1) | 4 (1) | 1 (25) |
| Status epilepticus | 4 (1) | 5 (1) | 1 (20) |
| Cor pulmonale | 3 (1) | 4 (1) | 2 (50) |
| Ruptured ectopic pregnancy | 3 (1) | 3 (1) | – |
| Fracture upper limb | 3 (1) | 3 (1) | 2 (67) |
| Endocarditis | 2 (0.4) | 2 (0.4) | 1 (33) |
| Pneumothorax | 2 (0.4) | 1 (0.2) | – |
| Hematothorax | 2 (0.4) | 3 (1) | 2 (50) |
| Rheumatic heart disease | – | 2 (0.4) | 1 (33) |
| Urinary Tract infection | – | 1 (0.2) | 1 (100) |
| Surgical site infection | – | 3 (1) | 2 (40) |

More than one diagnosis possible per patient: at discharge, 52 (11%) had one diagnosis, 91 (19%) 2 diagnoses, 55 (11%) 3 diagnoses, 293 (60%)>=4 diagnoses.

CAP-Community Acquired Pneumonia, COPD-Chronic Obstructive Pulmonary Disease.

[a]Death in the HDU; results are number and column percent of discharge diagnosis.

(N = 60, 12%), and stroke (n = 59, 12%), 24/59 (41%) being of hemorrhagic origin. Admission diagnoses to the HDU were broadly similar to those at discharge from HDU.

## Patient outcomes and predictors of mortality

Among the 491 patients who were admitted to the HDU, 146 (30%) died during their stay, while 36/345 (10%) died after being discharged. The most common deadliest conditions were sepsis, stroke, seizures, or aspiration pneumonia, with mortality rates of 51% to 65%, while mortality of patients with heart failure was 27% (Table 2).

The total number of in-hospital deaths was 182 (37%) and the overall 30-day mortality was 36% (177 deaths). A total of 25 out of 345 patients (7%) discharged from HDU were readmitted. The Kaplan-Meier survival curve in Fig 1 shows the probability of death over time. The median time to death after admission to the HDU was 2 days (IQR 1–5) and the likelihood of in-hospital mortality increased with prolonged hospital stay.

Predictors of in-hospital mortality shown in Table 3 were age ≥ 45 years versus 18–44 years (adjusted Hazard Ratio (aHR) 1.56, 95%CI 1.07–2.28, p = 0.03), hypotension (systolic blood pressure < 90 mmHg), (aHR 2.33, 95%CI 1.48–3.81, p < 0.001), GCS score ≤8 versus 14–15 (aHR 2.13, 95%CI 1.24–3.64, p = 0.02) and oxygen saturation at room air < 90% (aHR 1.62, 95%CI 1.04–2.51, p = 0.03). Patient medical history, reasons for admission, and source of admission were not significant predictors of in-hospital mortality.

## Association of clinical scores with in-hospital mortality

Validated clinical scores were used to assess the severity of illness and predict in-hospital mortality. The majority of admitted patients fulfilled the high-risk strata of these scores, underscoring the severity of their clinical conditions: using UVA, a total of 205 (42%), using NEWS, 297 (61%), using MEWS, 313 (65%), and using qSOFA 254 (52%) patients reached high risk scores, respectively (S1 Table). A high-risk NEWS score predicted in-hospital mortality with an odd ratio (OR) of 8.25

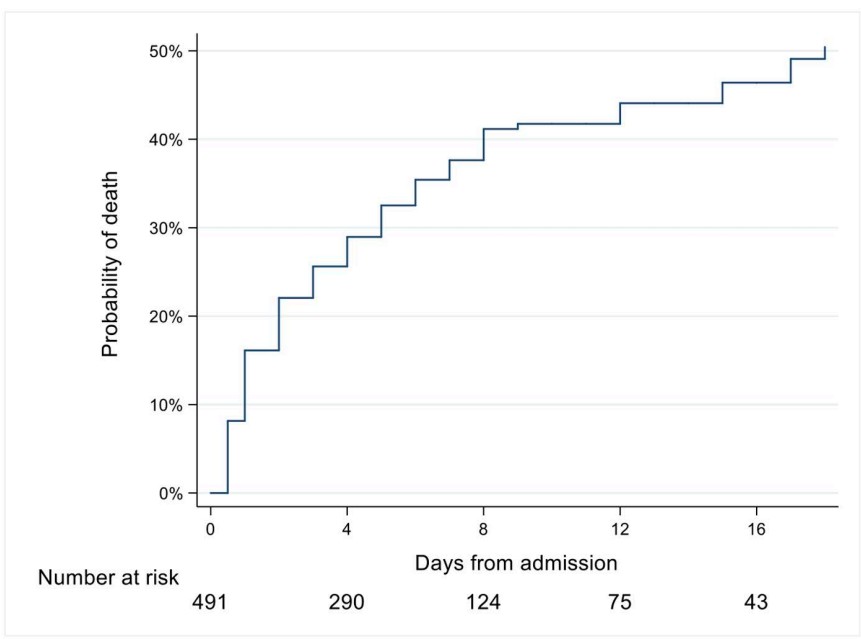

**Fig 1. In-hospital mortality rate among patients in the high-dependency unit.** The Kaplan-Meier curve shows the probability of death over time. The probabilities of death at 2, 6 and 8 days from admission were 22% (95% CI 19-26), 35% (95% CI 31-40) and 41% (95% CI 36-47), respectively.

**Table 3. Predictors of In-hospital mortality among patients in the high-dependency unit.**

| Characteristics | Univariable HR (95%CI)[a] | P value[a] | Multivariable HR (95%CI)[a,b] | P value[a,b] |
|---|---|---|---|---|
| Age categories (years) | | <0.001 | | 0.03 |
| <18 | 1.67 (0.88-3.19) | | 1.78 (0.81-3.93) | |
| 18–44 | Reference | | Reference | |
| ≥45 | 1.88 (1.37-2.58) | | 1.56 (1.07-2.28) | |
| Sex | | 0.32 | | 0.81 |
| Female | Reference | | Reference | |
| Male | 1.16 (0.87-1.55) | | 1.04 (0.74-1.46) | |
| Occupation | | 0.48 | | 0.46 |
| Non-farmer | Reference | | Reference | |
| Farmer | 1.14 (0.79-1.65) | | 1.19 (0.75-1.90) | |
| Admission from | | 0.004 | | 0.32 |
| Emergency Department | 1.18 (0.80-1.74) | | 1.20 (0.78-1.86) | |
| Theatre | 0.62 (0.38-1.01) | | 0.65 (0.30-1.42) | |
| Hospital ward | Reference | | Reference | |
| BMI (kg/m$^2$) | | 0.28 | | 0.09 |
| Underweight (BMI < 18.5) | 0.83 (0.49-1.40) | | 0.62 (0.33-1.15) | |
| Normal (BMI 18.5 −<25) | Reference | | Reference | |
| Overweight BMI (25 −<30) | 0.82 (0.55-1.21) | | 0.82 (0.54-1.25) | |
| Obesity (BMI ≥ 30) | 1.41 (0.87-2.29) | | 1.56 (0.89-2.72) | |
| GCS | | <0.001 | | 0.02 |
| ≤8 | 3.49 (2.45-4.97) | | 2.13 (1.24-3.64) | |
| 9−13 | 2.39 (1.65-3.49) | | 1.83 (1.08-3.08) | |
| 14-15 | Reference | | Reference | |
| Hypotension | 2.70 (1.79-4.07) | <0.001 | 2.33 (1.43-3.81) | <0.001 |
| Hypertension | 1.04 (0.77-1.41) | 0.79 | 1.07 (0.74-1.55) | 0.71 |
| Heart rate | | 0.08 | | 0.63 |
| ≤100/bpm | Reference | | Reference | |
| >100/bpm | 1.30 (0.97-1.74) | | 1.09 (0.77-1.53) | |
| Respiratory rate | | 0.11 | | 0.48 |
| ≤18/min | Reference | | Reference | |
| >18/min | 1.42 (0.91-2.22) | | 0.84 (0.51-1.38) | |
| Oxygen saturation on room air | | <0.001 | | 0.03 |
| ≥90% | Reference | | | |
| <90% | 2.90 (2.12-3.96) | | Reference | |
| Temperature (°C) | | 0.03 | 1.62 (1.04-2.51) | 0.63 |
| ≤37.8 | Reference | | Reference | |
| >37.8 | 1.69 (1.10-2.60) | | 0.89 (0.54-1.45) | |
| Medical history | | | | |
| HIV infection | 1.21 (0.73-1.99) | 0.48 | 1.44 (0.83-2.49) | 0.19 |
| Hypertension | 1.02 (0.73-1.42) | 0.92 | 0.67 (0.43-1.03) | 0.07 |
| Diabetes mellitus | 0.70 (0.45-1.07) | 0.08 | 0.89 (0.55-1.43) | 0.63 |
| Heart failure | 1.09 (0.60-2.01) | 0.78 | 0.94 (0.48-1.87) | 0.87 |
| Reason for admission | | | | |
| Trauma | 1.08 (0.64-1.80) | 0.78 | 0.95 (0.51-1.76) | 0.86 |

*(Continued)*

Table 3. (Continued)

| Characteristics | Univariable HR (95%CI)[a] | P value[a] | Multivariable HR (95%CI)[a,b] | P value[a,b] |
|---|---|---|---|---|
| Respiratory failure | 2.43 (1.78-3.32) | <0.001 | 1.38 (0.89-2.14) | 0.15 |
| Postoperative care | 0.57 (0.39-0.82) | <0.001 | 0.73 (0.33-1.60) | 0.43 |
| Symptoms and physical examinations | | | | |
| Airway problem | 2.46 (1.66-3.64) | <0.001 | 1.27 (0.79-2.05) | 0.32 |
| Breathing problem | 3.73 (2.20-6.32) | <0.001 | 1.47 (0.75-2.86) | 0.26 |
| Circulation problem | 2.28 (1.56-3.32) | <0.001 | 1.32 (0.83-2.08) | 0.24 |
| Disability problem | 3.01 (2.06-4.40) | <0.001 | 1.57 (0.89-2.77) | 0.12 |
| Exposure problem | 0.88 (0.65-1.19) | 0.40 | 1.08 (0.74-1.59) | 0.68 |

[a]Hazard Ratio (HR) and 95% Confident Intervals (CI) obtained from Cox regression.

[b]Adjusted for all variables shown in the table, missing data excluded, N = 487.

BP, blood pressure; Hypotension, systolic BP < 90mmHg; Hypertension, systolic BP ≥ 140 or diastolic BP ≥ 90; BMI, body mass index; GCS, Glasgow Coma Scale score.

(95%CI 4.34–15.67, p < 0.001). Fig 2 shows the AUROC analysis: The Area-Under-The-Curve (AUC) estimation revealed that UVA and NEWS scores had the highest predictive accuracy for in-hospital mortality, both with AUC values of 0.69 (95% CI 0.65–0.73) (Fig 2).

## Diagnostic imaging and treatment done during admission in the HDU

During the stay in the HDU, 460 diagnostic procedures were performed according to clinical indication including POCUS, echocardiography, lung ultrasound and ECG as a bedside investigation. Of these, 330 (72%) procedures were abnormal, as shown in S2 Table. The most common medication administered is shown in S3 Table. Antibiotics were used in 201 (58%), analgesics in 109 (32%), anti-hypertensives in 79 (23%) and heart failure medications in 68 (20%) patients. Vaso-active drugs were administered to 62 (13%) patients. Non-invasive ventilation was done in 25 (5%) patients

## Discussion

### Key findings

In this cohort study including all patients admitted within one year to a HDU of a regional referral hospital situated in rural Tanzania, most common diagnoses were sepsis, heart failure, and stroke, accompanied by hypertension, diabetes mellitus, acute kidney failure, and aspiration pneumonia. NCDs and their complications were the most relevant causes for HDU admission. Malaria accounted for 3% of diagnoses made upon discharge and was associated with a 54% mortality. Mortality was high with 30% of patients dying during HDU stay. While more than half of the patients with sepsis or stroke died in the HDU, the mortality of patients with decompensated heart failure was lower at 27%. Predictors of in-hospital mortality were age, a GCS score of ≤ 8, a systolic blood pressure of < 90 mmHg and an oxygen saturation of less than 90% at room air at admission. UVA, NEWS, MEWS, and qSOFA scores predicted in-hospital mortality with a moderate diagnostic accuracy.

### Comparison with other studies

Comparison of clinical information and outcomes from different critical care units is challenging, as settings and infrastructure are diverse. In our cohort, the median age of 46 years was higher than in another cohort of hospitalizes patients [38]. Since children are usually admitted to the pediatric department, only 6% of the patients admitted to the HDU were younger than 18 years. Sepsis was the most frequent diagnosis (20%), which is similar to studies from urban settings [16,17], but

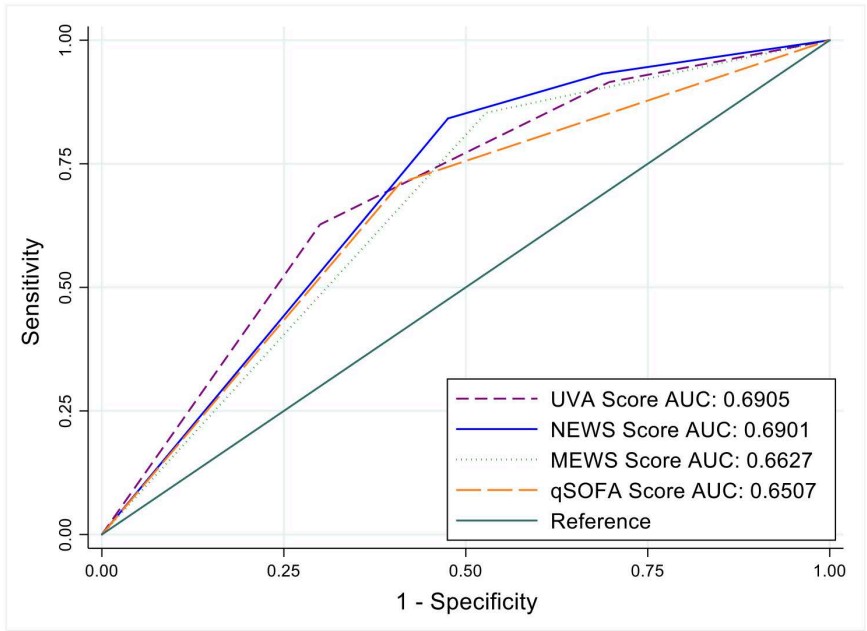

**Fig 2. Area under the receiver operating characteristic curves of clinical scores predicting in-hospital mortality of patients.** The area under the receiver operating characteristic curves (AUROC) shows the predictive accuracy of the Universal Vital Assessment (UVA) score, the National Early Warning Score (NEWS), the Modified Early Warning Score (MEWS), and the Quick Sequential Organ Failure Assessment (qSOFA) score for in-hospital mortality. The area under the curve (AUC) values were for UVA: 0.69 (95%CI 0.65–0.73); for NEWS: 0.69 (95%CI 0.65–0.73); for MEWS: 0.66 (95%CI 0.62–0.70); and for qSOFA: 0.65 (95%CI 0.61–0.69).

which is different to other studies where about 10% [15,28,39] to 44% [8] of patients had sepsis. The heterogeneity in the proportion of patients diagnosed with sepsis may be explained by different definitions used, and by different study designs such as retrospective study designs [15,28,39,40], or short prospective data collection periods of 2 days [8]. The majority of patients suffering from an NCD is in line with a recent large point prevalence study including patients from 22 African countries, where more than half of critically ill patients were admitted to the hospital because of an NCD [41]. The high number of patients with respiratory failure (N = 247, 50%) is in line with other African studies [12,15,39]. The underlying factors of respiratory failure are diverse and may be attributed to infectious or cardiac causes. Pulmonary edema leading to respiratory failure led to admission to HDU in 13%, which is higher than other studies (3–9%) [8,15,17], possibly due to poor control of underlying cardiopathies.

The high mortality rates in our study are consistent with those described in studies conducted in ICUs in Tanzania, Uganda, Malawi, and Ethiopia with mortality rates ranging between 25.6% and 52% in general [12,14–17,28,42] and 42–75% for patients with sepsis [40]. Comparison of outcomes with other studies is challenging due to different study designs [12,15,17,28,42], different periods of data assessment ranging from 6 weeks to 3 months [14,16] and because ICUs were situated in urban centres. The probability of death was highest during the first 5 days following admission which is similar to results reported in other studies [12,16,28]. In high-income countries, ICU-mortality has been reported to be 10.8% in the USA [43], while ICU- and in-hospital mortality was 19.1% and 23.9%, respectively, in an European multicenter cohort study [44].

Studies conducted in ICUs showed that the need for mechanical ventilation, the use of vasopressors or inotropes and an age > 45 years were associated with increased mortality rates [16,17,45]. In our study, hypotension, oxygen saturation < 90% and age > 45 years were strong predictors of in-hospital mortality.

The moderate prediction of mortality by scores is in line with other studies from Africa which included hospitalized patients in general wards [29,38,46]. In studies from high income countries, the NEWS score was the best predictor of in-hospital mortality compared with other scores, with an AUC of 0.87 [30,47]. Since people living with HIV are at greater risk of developing infections and sepsis, the UVA score is worth considering when stratifying mortality risk in Tanzania, where HIV prevalence is 4.7% [48]. All scores can be calculated based on clinical information only and are feasible in settings where laboratory results are not always available.

## Implications for practice

A leading cause of HDU admission in our study was stroke. In 65% of patients a CT could not be performed because of financial constraints of patients. Among patients who received a CT, haemorrhagic stroke was diagnosed in 41%. This might be explained due to the high proportion of patients with uncontrolled hypertension. Less than 15% of hypertensive people have controlled blood pressure in sub-Saharan Africa [49], and programes to screen for hypertension and to increase awareness of hypertension in the communities are needed.

Patients presented late in serious conditions. The severity of illness was demonstrated by a high-risk stratum of the scores at admission, which would have qualified patients to admission to an ICU, which is not implemented yet.

Late presentation and high mortality reflect systemic challenges in rural healthcare: The late presentation of patients in already serious conditions is due to lack of awareness of potentially life-threatening infectious – and non-communicable diseases in the communities, limited diagnostic tools to diagnose serious conditions and limited therapeutic options in the periphery, lack of transport, and the fact that a majority of patients do not have a health insurance to cover health-care costs. This highlights the importance of programmes to increase community awareness of diseases with potentially unfavorable outcomes and to strengthen prevention of NCDs, integrating diagnosis and treatment of NCDs into existing decentralized facilities such as HIV clinics [50], and training of health care personnel in diagnosing infectious diseases and NCDs in the periphery [51–53]. The implementation of referral systems and improving transport logistics to specialized centers is needed to maximise benefits from specialized care. While there are 35 ICU beds per 100'000 people in Germany [54], there is less than 1 ICU bed per 100'000 people in Africa [7,55]. One in eight patients in hospitals in Africa are critically ill, and two-thirds of them are cared for in general wards [41]. Therefore, implementation of more HDUs and ICUs is urgently needed in Africa. Training of healthcare personnel is needed to guarantee rapid and appropriate care for patients suffering from severe acute diseases or acute deterioration of pre-existing conditions. Moreover, access to health insurance needs to be ameliorated.

## Strengths

To the best of our knowledge, this is the first study reporting characteristics, diagnoses and outcomes of patients managed at a HDU in a hospital from a rural setting in sub-Saharan Africa. Strengths of this study are the prospective study design, the comprehensive data collection, the complete inclusion of all patients admitted to the HDU during one full calendar year, and the availability of the outcome in-hospital mortality for all patients, allowing reliable comparative analyses.

## Limitations

Our study has limitations. First, the access to diagnostic tools such as microbiological tests except Xpert MTB/RIF ULTRA®, HIV and malaria rapid tests, and laboratory tests beyond full blood picture, electrolytes, liver and renal function tests was limited. Therefore, diagnoses were mainly based on clinical and ultrasonographic findings, and some diseases could have been missed. However, all doctors working at the HDU were trained in clinical medicine, POCUS, focused echocardiography, and performance of ECG, and were supervised by experienced senior physicians who assessed all patients and reviewed all diagnoses. Second, although some of the treatments were administered before payment, financial constraints limited the use

of available diagnostic and therapeutic tools such as CT and hemodialysis. This might have a negative impact on patients outcomes. Third, 9% of patients were lost to follow-up after discharge from the hospital, and patients could not be followed for a longer period to determine their post discharge outcome. However, we could analyse all patients for the endpoint of in-hospital death. Fourth, non-invasive ventilation and the use of vasoactive drugs was implemented during the study period and would have been used more frequently if implemented earlier. Fifth, we could not determine the systemic inflammatory response syndrome (SIRS) score reliably, because white cell count was not available for all patients. Last, this was a single centre study, and generalisability of these findings to populations living in other settings might be limited.

## Conclusions

In conclusion, sepsis, heart failure, stroke, hypertension, diabetes mellitus, and renal failure were the most common diagnoses of patients admitted to a HDU of a referral hospital in rural Tanzania. While sepsis and stroke were among the deadliest conditions, patients with heart failure had a better prognosis. Scores such as UVA, MEWS, NEWS and qSOFA predicted in-Hospital mortality with a moderate accuracy. These findings underscore the need for improved critical care in low-resource rural settings. Decentralisation of critical care is needed to serve quickly patients in severe conditions. Future studies should explore interventions to reduce mortality in rural HDUs.

## Supporting information

**S1 Table. Association of clinical scores with in-hospital mortality among patients in the high-dependency unit.**
(DOCX)

**S2 Table. Diagnostic tests performed on patients in the high-dependency unit.**
(DOCX)

**S3 Table. Treatment and medication among patients in the high-dependency unit.**
(DOCX)

## Acknowledgments

We thank Fabian Fiechter, Sarah Adina Funk, and Albert Urwyler, University Hospital Basel, Switzerland; Wilson Lugano and Victor Zablon Urio, Benjamin Mkapa Hospital, Dodoma; and Vivienne Ayiana Mlawi, Jakaya Kikwete Cardiac Institute, Dar es Salaam, for training of the staff of the high dependency unit in critical care medicine. We thank Valentine Mteki, Ifakara Health Institute, for course organization.

## Author contributions

**Conceptualization:** Andrew Katende, Julie Rossier, Thomas Zoller, Maja Weisser, Martin Rohacek.

**Data curation:** Lulu Wilson, James Okuma.

**Formal analysis:** Lulu Wilson, James Okuma.

**Funding acquisition:** Julie Rossier, Martin Rohacek.

**Investigation:** Andrew Katende, Julie Rossier, Chipegwa Mlula, Christamonica Chitimbwa, Martin E. Mtula, Elibariki S. Mafole, Samuel S. Mutasingwa, Evance Mahundi, Mohamed K. Kipalaunga, Victor Myovela, Caspar Mbawala, Fanuel Faustine, Geofrey Mbunda, Winfrid Gingo, Faraja Kitila, Ipyana Mwasongwe, Claudia Bucher, Maja Weisser, Martin Rohacek.

**Methodology:** Andrew Katende, Julie Rossier, Chipegwa Mlula, Christamonica Chitimbwa, Lulu Wilson, James Okuma, Thomas Zoller, Maja Weisser, Martin Rohacek.

**Project administration:** Andrew Katende, Martin Rohacek.

**Resources:** Maja Weisser, Martin Rohacek.

**Software:** Lulu Wilson, Evance Mahundi.

**Supervision:** Daniel H. Paris, James Okuma, Maja Weisser, Martin Rohacek.

**Validation:** Martin Rohacek.

**Visualization:** James Okuma, Martin Rohacek.

**Writing – original draft:** Andrew Katende, Julie Rossier.

**Writing – review & editing:** Chipegwa Mlula, Christamonica Chitimbwa, Martin E. Mtula, Elibariki S. Mafole, Lulu Wilson, Samuel S. Mutasingwa, Evance Mahundi, Mohamed K. Kipalaunga, Victor Myovela, Caspar Mbawala, Fanuel Faustine, Geofrey Mbunda, Winfrid Gingo, Faraja Kitila, Ipyana Mwasongwe, Claudia Bucher, Daniel H. Paris, James Okuma, Thomas Zoller, Maja Weisser, Martin Rohacek.

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
