## [Decision Letter · Decision Letter 0]

Dear Dr. Rohacek,

Thank you for submitting your manuscript to PLOS ONE. After careful consideration, we feel that it has merit but does not fully meet PLOS ONE’s publication criteria as it currently stands. Therefore, we invite you to submit a revised version of the manuscript that addresses the points raised during the review process.

We look forward to receiving your revised manuscript.

Kind regards,

Abraham Aregay Desta, MSc, PhD candidate.

Academic Editor

PLOS ONE

Journal Requirements:

2. Thank you for stating the following financial disclosure: [Martin Rohacek received fund from the Else Kröner Fresenius Foundation, Germany, Julie Rossier received funds from the Freiwillige Akatremische Gesellschaft Basel, Switzerland]. 

Reviewers' comments:

Reviewer's Responses to Questions

**Comments to the Author**

1. Is the manuscript technically sound, and do the data support the conclusions?

Reviewer #1: Yes

Reviewer #2: Partly

2. Has the statistical analysis been performed appropriately and rigorously?

Reviewer #1: Yes

Reviewer #2: Yes

3. Have the authors made all data underlying the findings in their manuscript fully available?

Reviewer #1: No

Reviewer #2: Yes

4. Is the manuscript presented in an intelligible fashion and written in standard English?

Reviewer #1: Yes

Reviewer #2: Yes

Reviewer #1: The study addresses an important gap in the literature by providing the first detailed description of diagnoses and outcomes in a rural high dependency unit (HDU) in sub-Saharan Africa.

The methodology is robust, with a prospective cohort design, clear inclusion/exclusion criteria, and comprehensive data collection.

The findings are significant, highlighting high mortality rates and predictors of in-hospital mortality, which have important implications for critical care in low-resource settings.

Areas for Improvement:

Clarity and Structure: Some sections (e.g., Results, Discussion) could be streamlined for better readability.

Contextualization: The paper would benefit from a stronger emphasis on the broader implications of the findings for policy and practice in rural healthcare.

Limitations: While limitations are acknowledged, they could be discussed in more depth, particularly regarding the impact of financial constraints and diagnostic limitations on patient outcomes.

Language: Minor grammatical and stylistic improvements are needed to enhance clarity and flow.

Overall Assessment:

The paper is scientifically sound and makes a valuable contribution to the field. The required revisions are relatively minor and primarily focus on improving clarity, structure, and contextualization.

Reviewer #2: The study addresses an important gap in understanding critically ill patient outcomes in rural HDUs, making it highly relevant for clinicians and policymakers. Here are my recommendations:

1. The introduction focuses heavily on HDU setup rather than framing a clear research gap.

2. The objective should be explicitly stated earlier in the introduction and aligned more clearly with the research question.

3. The methods section needs more focus on patient selection,sample size, data collection, and statistical analysis rather than HDU implementation details.

4. The criteria for selecting variables in multivariable analysis are not clearly stated. Providing justification based on univariate p-values, clinical relevance, or confounding adjustment would improve transparency.

5. There are inconsistencies in reporting p-values and confidence intervals. Using a uniform format (e.g., p = 0.03, 95% CI 1.07–2.28) would enhance readability.

6. Realign all the segments to the objectives rather and HDU implementation.

**Do you want your identity to be public for this peer review?** For information about this choice, including consent withdrawal, please see our Privacy Policy

Reviewer #1: **Yes: ** Yamlak Gindola

Reviewer #2: No

---

## [Author Response · Author response to Decision Letter 1]

10 Apr 2025

To April 10th 2025

Dr Abraham Aregay Desta

Editor

PLOS one

Dear Dr Desta

We thank you and the reviewers very much for the review of our manuscript

PONE-D-25-05201

"Diagnoses and outcomes of critically ill patients admitted to a high dependency unit of a rural referral hospital in Tanzania: A Prospective Cohort Study"

We changed the title to “First Insights into Critical Care Outcomes in a Rural Tanzanian High Dependency Unit: A prospective cohort study” as suggested by a reviewer.

We revised the manuscript according to the valuable comments of the reviewers and responded in a point-by-point reply below.

We uploaded a clean version of the manuscript and a version with track changes.

We uploaded the minimal dataset to https://zenodo.org/records/15177213

We checked all statistical analyses by running them again and corrected 2 numbers in Table 1 and one number in Table 3 which is marked in yellow.

Martin Rohacek received funds from the Else Kröner Fresenius Foundation, Germany, Julie Rossier received funds from the Freiwillige Akademische Gesellschaft Basel, Switzerland. The funders had no role in study design, data collection and analysis, decision to publish, or preparation of the manuscript.

Thank you very much for evaluating our revised version of the manuscript.

Sincerely yours

PD Dr med Martin Rohacek

Swiss TPH

Kreuzstrasse 2

4123 Allschwil

Switzerland

University of Basel, Switzerland

Ifakara Health Institute, Tanzania

martin.rohacek@swisstph.ch, mrohacek@ihi.or.tz

Point By Point reply to reviewers’ comments which was attached to the e-mail from March 19th and April 1st

A Review for Manuscript Number PONE-D-25-05201

"Diagnoses and outcomes of critically ill patients admitted to a high dependency unit of a rural referral hospital in Tanzania: A prospective cohort study"

Title

Current Title: Diagnoses and outcomes of critically ill patients admitted to a high dependency unit of a rural referral hospital in Tanzania: A prospective cohort study

Comments:

Consider making the title more concise. For example:

"Critical Care Outcomes in a Rural Tanzanian High Dependency Unit: A Prospective Cohort Study"

Alternatively, emphasize the novelty of the study:

"First Insights into Critical Care Outcomes in a Rural Tanzanian High Dependency Unit: A Prospective Cohort Study"

Response:

We thank the reviewer for this comment. We changed the title to "First Insights into Critical Care Outcomes in a Rural Tanzanian High Dependency Unit: A Prospective Cohort Study" as suggested

Abstract

Comments:

Background: Briefly mention the gap in literature (e.g., lack of data on HDUs in rural sub-Saharan Africa).

Response:

We thank the reviewer for this comment. We revised the background of the abstract as follows:

Background: Data on high dependency units (HDU) situated in rural sub-Saharan Africa are lacking. In this study, we describe patient’s characteristics, diagnoses, and outcomes of critically ill patients admitted to a high dependency unit of a referral hospital in rural Tanzania, and factors associated with in-hospital mortality.

Methods:

Comment:

Clarify the sample size earlier (e.g., “491 patients admitted to the HDU”).

Response: We included this information into the first sentence of the methods section on page 3.

Results:

Comment:

Highlight the most striking finding (e.g., “Mortality during HDU stay was 30%, with sepsis and stroke being the deadliest conditions”).

Response:

We thank the reviewer for this comment. We moved the section “Patient outcomes and mortality” up, adapted the numbers of S1-S3 Table and added the sentence “The most common deadliest conditions were sepsis, stroke, seizures, or aspiration pneumonia, with mortality rates of 51 to 65%, while mortality of patients with heart failure was 27% on page 11.

Conclusion:

Comment:

Add a sentence on implications for policy or practice (e.g., “These findings underscore the need for improved critical care resources in rural settings”).

Response:

We thank the reviewer for this suggestion, which we added to the conclusion.

Introduction

Comment:

Contextualize the Problem: Add a sentence on the burden of non-communicable diseases (NCDs) in Tanzania, as they are a major focus of the study.

Response:

We thank the reviewer for this comment.

We added information about NCDs to the first sentence of the introduction on page 3: Critical care medicine is important to manage seriously ill patients suffering from sepsis, pneumonia, and from non-communicable diseases (NCDs) such as heart failure and stroke. Globally, NCDs killed at least 43 million people in 2021, and 73% occurred in low- and middle-income countries.

Comment:

Justify the Study: Emphasize why this study is novel (e.g., “This is the first study to describe outcomes in a rural HDU in sub-Saharan Africa”).

Response:

We added the following sentence at the end of the Introduction: “This is the first study on outcomes of patients admitted to a HDU in rural sub-Saharan Africa”. On page 3

Comment:

Clarify Objectives: Rephrase the objectives to be more specific (e.g., “To describe patient characteristics, diagnoses, and outcomes, and identify predictors of in-hospital mortality”).

Response:

We rephrased the objectives: “The objectives of this study were to describe characteristics, diagnoses, and outcomes of critically ill patients admitted to a recently implemented HDU of a referral hospital in rural Tanzania, and to describe predictors of in-hospital mortality” in the Introduction section on page 3 last 4 lines

Methods

Comment:

Study Design: Clarify the prospective nature of the study earlier in the section.

Response:

We thank the reviewer for this comment.

We included this into the first sentence in the methods section “This prospective observational single center cohort study including 491 patients was conducted at the HDU of the St. Francis Regional Referral Hospital (SFRRH), Ifakara, Tanzania” on page 3

Comment:

Data Collection: Provide more detail on how data quality was ensured (e.g., training of data collectors, use of standardized forms).

Response:

We added the following sentence in the section study procedures and data collection on page 6:

“Data were collected by clinicians working at the HDU, and responded to queries raised by the data manager and the statistician who cleaned the data. Before the start of the study, all members were trained and instructed how to fill data into the standardized electronic data collection tools”

Comment:

Statistical Analysis: Briefly explain why specific statistical methods were chosen (e.g., Cox regression for mortality predictors).

Response:

We thank the reviewer for the suggestion, we have revised the Statistical Analysis section to clarify the rationale for the selected methods. The updated section now explicitly states why each statistical approach was chosen:

• Descriptive statistics were used to summarize baseline characteristics and diagnoses, as they provide a clear and concise overview of patient demographics and clinical profiles.

• Kaplan-Meier survival curves were used to visualize mortality rates over time, which is an effective method for estimating survival probabilities.

• Cox proportional hazards regression was utilized to identify factors associated with in-hospital mortality. This method was chosen because it accounts for varying follow-up times among patients and provides adjusted hazard ratios, which allow for meaningful comparisons of risk factors.

• AUROC analysis was performed to evaluate the predictive accuracy of existing mortality scores, as it is a well-established method for assessing the discriminative ability of clinical prediction models.

These statistical approaches were selected to ensure that our findings are both robust and clinically meaningful. We have incorporated these revisions into the manuscript.

Revised Statistical Analysis Section (for Manuscript)

Statistical Analysis

Descriptive statistics were used to summarize baseline characteristics and diagnoses at admission and discharge. Kaplan-Meier survival curves were employed to estimate and visualize mortality rates over time. The association between clinical factors, risk scores, and mortality was assessed using Cox proportional hazards regression, which accounts for differences in follow-up time and provides adjusted hazard ratios to quantify the strength of associations. To evaluate the predictive performance of early warning scores, we conducted area under the receiver operating characteristic curve (AUROC) analysis, which measures the discriminative ability of each score in predicting in-hospital mortality. All statistical analyses were performed using Stata version 16. A p-value of <0.05 was considered statistically significant.

Revised Methods Section in Abstract (for Manuscript)

Descriptive analyses were performed, followed by univariate and multivariate modeling to identify predictors of in-hospital mortality. The area under the receiver operating characteristic (ROC) curve was used to assess the predictive accuracy of early warning scores.

Comment:

Ethics: Mention how oral consent was documented to address potential concerns about consent validity.

Response:

We thank the reviewer for this comment.

Written informed consent was waived by both ethic committees. If patients were conscious, they were informed that data would be used for research purposes. In case of unconsciousness, relatives were informed. However, it was not specifically documented if the patient or the relative was informed.

Results

Comments:

Clarify Key Findings: Highlight the most important results in the text (e.g., “Sepsis and stroke were associated with the highest mortality rates”).

Response:

Thank you for this comment, the section about outcomes was moved up to page 11, and the sentence “The most common deadliest conditions were sepsis, stroke, seizures, or aspiration pneumonia, with mortality rates of 51 to 65%, while mortality of patients with heart failure was 27%” was added at the beginning of the section.

Comment:

Simplify Tables: Consider merging or simplifying tables to improve readability (e.g., combine Tables 1 and 2).

Response:

We thank the reviewer for this comment. We simplified Table 1 and 2, however, we could not merge Table 1 and 2 together because table 1 shows baseline characteristics and Table 2 shows results

Comment:

Visuals: Ensure figures (e.g., ROC curves) are clearly labeled and interpretable.

Response:

We revised the labels and legends of the figures

Discussion

Comment:

Structure: Use subheadings (e.g., “Key Findings”, “Comparison with Literature”, “Implications for Practice”) to improve flow.

Response: We added subheadings in the discussion section.

Comment:

Contextualize Findings: Discuss how the high mortality rates reflect systemic challenges in rural healthcare.

Response:

Thank you for this comment, we added this on page 16: “Late presentation and high mortality reflect systemic challenges in rural healthcare: The late presentation of patients in already serious conditions is due to lack of awareness of potentially life-threatening infectious – and non-communicable diseases in the communities, limited diagnostic and therapeutical options to diagnose serious conditions in the periphery, lack of transport, and the fact that a majority of patients do not have a health insurance to cover the costs”.

Comment:

Policy Implications: Add a paragraph on how the findings can inform policy (e.g., “These results highlight the need for increased investment in rural critical care infrastructure”).

Response:

Thank you for this comment, we added this on page 16:

This highlights the importance of programmes to increase community awareness of diseases with potentially unfavourable outcomes, training of health care personnel in diagnosing infectious diseases and NCDs, implementation of referral systems, strengthening transport to specialized centers, implementing emergency departments and critical care facilities, and training of healthcare personnel in emergency medicine and critical care. Moreover, access to health insurance needs to be ameliorated, to guarantee rapid and appropriate care for patients suffering from severe acute diseases or acute deterioration of pre-existing conditions.

Comment:

Limitations: Expand on the impact of lost-to-follow-up cases and diagnostic limitations.

Response:

Thank you for this comment, we expanded this in the limitation section, page 17

Third, 9% of patients were lost to follow-up after discharge from the hospital, and patients could not be followed for a longer period to determine their post discharge outcome. However, we could analyse all patients for the endpoint of in-hospital death.

Conclusion

Comment:

Call to Action: Add a sentence on next steps (e.g., “Future studies should explore interventions to reduce mortality in rural HDUs”).

Response: Thank you for this comment, We added this sentence at the end.

Comment:

Broader Impact: Mention how the study contributes to global health equity (e.g., “This study provides critical insights into improving healthcare access in low-resource settings”).

Response:

We added “These findings underscore the need for improved critical care in low-resource rural settings

References

Comment:

Ensure all references are formatted consistently according to the journal’s guidelines.

Response:

We checked all references and formatted accordingly

Comment:

Include more references from sub-Saharan Africa to strengthen the regional context.

Response:

We added the following References in the discussion section

Beaney T, Burrell LM, Castillo RR, et al. May Measurement Month 2018: a pragmatic global screening campaign to raise awareness of blood pressure by the International Society of Hypertension. Eur Heart J 2019;40:2006-2017. doi: 10.1093/eurheartj/ehz300

WHO PEN and integrated outpatient care for severe, chronic NCDS at first referral hospitals in the African region (PEN-PLUS). 2019.

https://www.afro.who.int/publications/who-pen-and-integrated-outpatient-care-severe-chronic-ncds-first-referral-hospitals. Last access March 24th 2025

Klassen SL, Okello E, Ferrer JME, et al. Decentralization and Integration of Advanced Cardiac Care for the World's Poorest Billion Through the PEN-Plus Strategy for Severe Chronic Non-Communicable Disease. Glob Heart 2024;19:33. doi: 10.5334/gh.1313

WHO. HEARTS; Technical package for cardiovascular disease management in primary health care: Tool for the development of a consensus protocol for treatment of hypertension: technical package for cardiovascular disease management in primary health care; https://www.who.int/publications/i/item/WHO-NMH-NVI-19-8. Last access March 24th 2025

Kivuyo S, Birungi J, Okebe J, et al. Integrated management of HIV, diabetes, and hypertension in sub-Saharan Africa (INTE-AFRICA): a pragmatic cluster-randomised, controlled trial. Lancet 2023;402:1241-1250. doi: 10.1016/S0140-6736(23)01573-8

Overall Suggestions

Language and Clarity:

Comment:

• Simplify complex sentences for better readability.

• Avoid jargon and define acronyms (e.g., HDU, NEWS, qSOFA) at first use.

Response:

We checked the manuscript and revised appropriately.

Figures and Tables:

Comment:

• Ensure all figures and tables are high-quality and clearly labeled.

• Add a brief narrative summary for each table/figure in the text.

Response:

All Figures were checked and brief narrative summary was added to each table and figure in the text

Data Availability:

Comment:

• Clarify how readers can access the data (e.g., repository name, DOI).

Response:

We stated that “The datasets used and/or analysed during the current study are available from the corresponding author on reasonable request” on page 17.

Strengths and Limitations:

Comment:

• Highlight the study’s strengths (e.g., prospective design, comprehensive data collection) more prominently.

Response:

We added this to the “Strengths Section” on page 16

Comment:

• Discuss limitations in more depth (e.g., impact of financial constraints on patient outcomes).

Response:

We added “ Second, fin

---

## [Editor Report · Decision Letter 1]

Please try to indicate the line number and page while you respond for each commentConsider modifying your title as “Diagnoses and Critical Care Outcomes in a Rural Tanzanian High Dependency Unit: A Prospective Cohort Study”. Because, despite the “first insights” draws attention, it can lead a bit vague and may loss academic tone. In addition, this title should directly be coherent with the background, objectives and the results as well, you have showed most common diagnoses.Can you please provide the justification why you conducted multivariable analysis, as you have no sample (you have included all the study participants), or to whom are you going to infer?In the statistical analysis part for the association between clinical factors, risk scores, and mortality was assessed using Cox proportional hazards regression. However, there is no information how you checked the proportional hazard assumption. You will have meaningful comparisons of risk factors if the assumptions are held.Better to replace describe by identify in the “to describe predictors of in-hospital mortality”You have mentioned that Kaplan-Meier survival curves to visualize mortality rate s over time in your feedback however, there is any description about this in your statistical analysis. Additionally, if you used Kaplan-Meier survival curves how do check the effect of the censoring pattern or how do you check the homogeneity of the groups?Overall, you must show the methods clearly, especially your statistical analysis is not clear and try to improve this section.Improve your discussion and compare it with some international findings and show the gaps clearly. Rember that you have mentioned “Critical care services in sub-Saharan Africa increased since the COVID-19 pandemic, but remain limited compared to high-income countries” in the introduction part

We look forward to receiving your revised manuscript.

Kind regards,

Abraham Aregay Desta 

Academic Editor

PLOS ONE

Additional Editor Comments:

Dear Authors,

Thank you for submitting your revised manuscript entitled “First Insights into Critical Care Outcomes in a Rural Tanzanian High Dependency Unit: A Prospective Cohort Study” (Manuscript ID: PONE-D-25-05201R1) to PLOS ONE.

After careful evaluation of your submission, we appreciate the important contribution your study offers in understanding critical care delivery in a low-resource and rural setting. However, the manuscript still requires major revisions before it can be considered for publication.

Below are the primary concerns and suggestions that must be addressed:

Please try to indicate the line number and page while you respond for each comment

Consider modifying your title as “Diagnoses and Critical Care Outcomes in a Rural Tanzanian High Dependency Unit: A Prospective Cohort Study”. Because, despite the “first insights” draws attention, it can lead a bit vague and may loss academic tone. In addition, this title should directly be coherent with the background, objectives and the results as well, you have showed most common diagnoses.

Can you please provide the justification why you conducted multivariable analysis, as you have no sample (you have included all the study participants), or to whom are you going to infer?

In the statistical analysis part for the association between clinical factors, risk scores, and mortality was assessed using Cox proportional hazards regression. However, there is no information how you checked the proportional hazard assumption. You will have meaningful comparisons of risk factors if the assumptions are held.

Better to replace describe by identify in the “to describe predictors of in-hospital mortality”

You have mentioned that Kaplan-Meier survival curves to visualize mortality rate s over time in your feedback however, there is any description about this in your statistical analysis. Additionally, if you used Kaplan-Meier survival curves how do check the effect of the censoring pattern or how do you check the homogeneity of the groups?

Overall, you must show the methods clearly, especially your statistical analysis is not clear and try to improve this section.

Improve your discussion and compare it with some international findings and show the gaps clearly. Rember that you have mentioned “Critical care services in sub-Saharan Africa increased since the COVID-19 pandemic, but remain limited compared to high-income countries” in the introduction part

Please revise the manuscript based on the above feedbacks. We also encourage you to include a point-by-point response addressing each concern raised. Highlight all changes in the revised manuscript for easier evaluation.

Once we receive your revised submission, we will proceed with further editorial review. We appreciate your efforts in advancing research in global health and look forward to receiving your revised manuscript.

Sincerely,

Abraham Desta

Academic Editor

PLOS ONE

---

## [Author Response · Author response to Decision Letter 2]

17 Apr 2025

Point By Point reply to reviewers’ comments which was attached to the e-mail from March 19th and April 1st

A Review for Manuscript Number PONE-D-25-05201

Title

Current Title: "Diagnoses and Critical Care Outcomes in a rural Tanzanian High Dependency Unit: A Prospective Cohort Study"

Comments:

Consider making the title more concise. For example:

"Critical Care Outcomes in a Rural Tanzanian High Dependency Unit: A Prospective Cohort Study"

Alternatively, emphasize the novelty of the study:

"First Insights into Critical Care Outcomes in a Rural Tanzanian High Dependency Unit: A Prospective Cohort Study"

Response:

We thank the reviewer for this comment. We changed the title to "First Insights into Critical Care Outcomes in a Rural Tanzanian High Dependency Unit: A Prospective Cohort Study" as suggested, but changed the title to "Diagnoses and Critical Care Outcomes in a rural Tanzanian High Dependency Unit: A Prospective Cohort Study" later.

Abstract

Comments:

Background: Briefly mention the gap in literature (e.g., lack of data on HDUs in rural sub-Saharan Africa).

Response:

We thank the reviewer for this comment. We revised the background of the abstract as follows:

Background: Data on high dependency units (HDU) situated in rural sub-Saharan Africa are lacking. In this study, we describe patient’s characteristics, diagnoses, and outcomes of critically ill patients admitted to a high dependency unit of a referral hospital in rural Tanzania, and factors associated with in-hospital mortality (page 2, line 43 – 46)

Methods:

Comment:

Clarify the sample size earlier (e.g., “491 patients admitted to the HDU”).

Response: We included this information into the first sentence of the methods section on page 3, line 107.

Results:

Comment:

Highlight the most striking finding (e.g., “Mortality during HDU stay was 30%, with sepsis and stroke being the deadliest conditions”).

Response:

We thank the reviewer for this comment. We moved the section “Patient outcomes and mortality” up to page 11 line 308 ff, adapted the numbers of S1-S3 Table and added the sentence “The most common deadliest conditions were sepsis, stroke, seizures, or aspiration pneumonia, with mortality rates of 51 to 65%, while mortality of patients with heart failure was 27% on page 11 line 310-312.

Conclusion:

Comment:

Add a sentence on implications for policy or practice (e.g., “These findings underscore the need for improved critical care resources in rural settings”).

Response:

We thank the reviewer for this suggestion, which we added to the conclusion on page 18 line 525-526.

Introduction

Comment:

Contextualize the Problem: Add a sentence on the burden of non-communicable diseases (NCDs) in Tanzania, as they are a major focus of the study.

Response:

We thank the reviewer for this comment.

We added information about NCDs to the first sentence of the introduction on page 3 line 78-81: Critical care medicine is important to manage seriously ill patients suffering from sepsis, pneumonia, and from non-communicable diseases (NCDs) such as heart failure and stroke. Globally, NCDs killed at least 43 million people in 2021, and 73% occurred in low- and middle-income countries.

Comment:

Justify the Study: Emphasize why this study is novel (e.g., “This is the first study to describe outcomes in a rural HDU in sub-Saharan Africa”).

Response:

We added the following sentence at the end of the Introduction: “This is the first study on outcomes of patients admitted to a HDU in rural sub-Saharan Africa”. On page 3 line 102-103.

Comment:

Clarify Objectives: Rephrase the objectives to be more specific (e.g., “To describe patient characteristics, diagnoses, and outcomes, and identify predictors of in-hospital mortality”).

Response:

We rephrased the objectives: “The objectives of this study were to describe characteristics, diagnoses, and outcomes of critically ill patients admitted to a recently implemented HDU of a referral hospital in rural Tanzania, and to describe predictors of in-hospital mortality” in the Introduction section on page 3 line 99-102.

Methods

Comment:

Study Design: Clarify the prospective nature of the study earlier in the section.

Response:

We thank the reviewer for this comment.

We included this into the first sentence in the methods section “This prospective observational single center cohort study including 491 patients was conducted at the HDU of the St. Francis Regional Referral Hospital (SFRRH), Ifakara, Tanzania” on page 3 line 107-108.

Comment:

Data Collection: Provide more detail on how data quality was ensured (e.g., training of data collectors, use of standardized forms).

Response:

We added the following sentence in the section study procedures and data collection on page 6 line 189-192:

“Data were collected by clinicians working at the HDU, and responded to queries raised by the data manager and the statistician who cleaned the data. Before the start of the study, all members were trained and instructed how to fill data into the standardized electronic data collection tools”

Comment:

Statistical Analysis: Briefly explain why specific statistical methods were chosen (e.g., Cox regression for mortality predictors).

Response:

We thank the reviewer for the suggestion, we have revised the Statistical Analysis section to clarify the rationale for the selected methods. The updated section now explicitly states why each statistical approach was chosen:

• Descriptive statistics were used to summarize baseline characteristics and diagnoses, as they provide a clear and concise overview of patient demographics and clinical profiles.

• Kaplan-Meier survival curves were used to visualize mortality rates over time, which is an effective method for estimating survival probabilities.

• Cox proportional hazards regression was utilized to identify factors associated with in-hospital mortality. This method was chosen because it accounts for varying follow-up times among patients and provides adjusted hazard ratios, which allow for meaningful comparisons of risk factors.

• AUROC analysis was performed to evaluate the predictive accuracy of existing mortality scores, as it is a well-established method for assessing the discriminative ability of clinical prediction models.

These statistical approaches were selected to ensure that our findings are both robust and clinically meaningful. We have incorporated these revisions into the manuscript.

We revised the statistical analysis section accordingly on page 7 line 225 to 245.

Comment:

Ethics: Mention how oral consent was documented to address potential concerns about consent validity.

Response:

We thank the reviewer for this comment.

Written informed consent was waived by both ethic committees. If patients were conscious, they were informed that data would be used for research purposes. In case of unconsciousness, relatives were informed. However, it was not specifically documented if the patient or the relative was informed (page 7 line 248-252).

Results

Comments:

Clarify Key Findings: Highlight the most important results in the text (e.g., “Sepsis and stroke were associated with the highest mortality rates”).

Response:

Thank you for this comment, the section about outcomes was moved up to page 11 line 308ff, and the sentence “The most common deadliest conditions were sepsis, stroke, seizures, or aspiration pneumonia, with mortality rates of 51 to 65%, while mortality of patients with heart failure was 27%” was added at the beginning of the section page 11 line 310-312.

Comment:

Simplify Tables: Consider merging or simplifying tables to improve readability (e.g., combine Tables 1 and 2).

Response:

We thank the reviewer for this comment. We simplified Table 1 and 2, however, we could not merge Table 1 and 2 together because table 1 shows baseline characteristics and Table 2 shows results

Comment:

Visuals: Ensure figures (e.g., ROC curves) are clearly labeled and interpretable.

Response:

We revised the labels and legends of the figures.

Discussion

Comment:

Structure: Use subheadings (e.g., “Key Findings”, “Comparison with Literature”, “Implications for Practice”) to improve flow.

Response: We added subheadings in the discussion section.

Comment:

Contextualize Findings: Discuss how the high mortality rates reflect systemic challenges in rural healthcare.

Response:

Thank you for this comment, we added this on page 16 line 465-477: “Late presentation and high mortality reflect systemic challenges in rural healthcare: The late presentation of patients in already serious conditions is due to lack of awareness of potentially life-threatening infectious – and non-communicable diseases in the communities, limited diagnostic and therapeutical options to diagnose serious conditions in the periphery, lack of transport, and the fact that a majority of patients do not have a health insurance to cover the costs”.

Comment:

Policy Implications: Add a paragraph on how the findings can inform policy (e.g., “These results highlight the need for increased investment in rural critical care infrastructure”).

Response:

Thank you for this comment, we added more information on page 16 line 477-488.

Comment:

Limitations: Expand on the impact of lost-to-follow-up cases and diagnostic limitations.

Response:

Thank you for this comment, we expanded this in the limitation section, page 17 line 507-509:

“Third, 9% of patients were lost to follow-up after discharge from the hospital, and patients could not be followed for a longer period to determine their post discharge outcome. However, we could analyse all patients for the endpoint of in-hospital death”.

Conclusion

Comment:

Call to Action: Add a sentence on next steps (e.g., “Future studies should explore interventions to reduce mortality in rural HDUs”).

Response: Thank you for this comment, We added this sentence on page 18 line 527.

Comment:

Broader Impact: Mention how the study contributes to global health equity (e.g., “This study provides critical insights into improving healthcare access in low-resource settings”).

Response:

We added “These findings underscore the need for improved critical care in low-resource rural settings on page 18 line 525-526.

References

Comment:

Ensure all references are formatted consistently according to the journal’s guidelines.

Response:

We checked all references and formatted accordingly

Comment:

Include more references from sub-Saharan Africa to strengthen the regional context.

Response:

We added the following References in the discussion section

Beaney T, Burrell LM, Castillo RR, et al. May Measurement Month 2018: a pragmatic global screening campaign to raise awareness of blood pressure by the International Society of Hypertension. Eur Heart J 2019;40:2006-2017. doi: 10.1093/eurheartj/ehz300

WHO PEN and integrated outpatient care for severe, chronic NCDS at first referral hospitals in the African region (PEN-PLUS). 2019.

https://www.afro.who.int/publications/who-pen-and-integrated-outpatient-care-severe-chronic-ncds-first-referral-hospitals. Last access March 24th 2025

Klassen SL, Okello E, Ferrer JME, et al. Decentralization and Integration of Advanced Cardiac Care for the World's Poorest Billion Through the PEN-Plus Strategy for Severe Chronic Non-Communicable Disease. Glob Heart 2024;19:33. doi: 10.5334/gh.1313

WHO. HEARTS; Technical package for cardiovascular disease management in primary health care: Tool for the development of a consensus protocol for treatment of hypertension: technical package for cardiovascular disease management in primary health care; https://www.who.int/publications/i/item/WHO-NMH-NVI-19-8. Last access March 24th 2025

Kivuyo S, Birungi J, Okebe J, et al. Integrated management of HIV, diabetes, and hypertension in sub-Saharan Africa (INTE-AFRICA): a pragmatic cluster-randomised, controlled trial. Lancet 2023;402:1241-1250. doi: 10.1016/S0140-6736(23)01573-8

Baker T, Scribante J, Elhadi M, Ademuyiwa A, Osinaike B, Owoo C, et al. The African Critical Illness Outcomes Study (ACIOS): a point prevalence study of critical illness in 22 nations in Africa. The Lancet. 2025;405: 715–724. doi:10.1016/S0140-6736(24)02846-0

Overall Suggestions

Language and Clarity:

Comment:

• Simplify complex sentences for better readability.

• Avoid jargon and define acronyms (e.g., HDU, NEWS, qSOFA) at first use.

Response:

We checked the manuscript and revised appropriately.

Figures and Tables:

Comment:

• Ensure all figures and tables are high-quality and clearly labeled.

• Add a brief narrative summary for each table/figure in the text.

Response:

All Figures were checked and brief narrative summary was added to each table and figure in the text

Data Availability:

Comment:

• Clarify how readers can access the data (e.g., repository name, DOI).

Response:

We added “The datasets used and/or analysed during the current study are available under https://zenodo.org/records/15177213 with restricted access” on page 18 line 533-53

Strengths and Limitations:

Comment:

• Highlight the study’s strengths (e.g., prospective design, comprehensive data collection) more prominently.

Response:

We added this to the “Strengths Section” on page 17 line 491-496

Comment:

• Discuss limitations in more depth (e.g., impact of financial constraints on patient outcomes).

Response:

We added “ Second, financial constraints limited the use of available diagnostic and therapeutic tools such as CT and hemodialysis. This might have a negative impact on patients outcomes.” To the limitation section” on page 17 line 504-507.

Point by Point reply to additional points raised in the e-mail from April 1st 2025

5. Review Comments to the Author

Reviewer #1:

Comments:

The study addresses an important gap in the literature by providing the first detailed description of diagnoses and outcomes in a rural high dependency unit (HDU) in sub-Saharan Africa.

The methodology is robust, with a prospective cohort design, clear inclusion/exclusion criteria, and comprehensive data collection.

The findings are significant, highlighting high mortality rates and predictors of in-hospital mortality, which have important implications for critical care in low-resource settings.

Response:

We thank the reviewer very much for this positive comments

Areas for Improvement:

Comments:

Clarity and Structure: Some sections (e.g., Results, Discussion) could be streamlined for better readability.

Contextualization: The paper would benefit from a stronger emphasis on the broader implications of the findings for policy and practice in rural healthcare.

Limitations: While limitations are acknowledged, they could be discussed in more depth, particularly regarding the impact of financial constraints and diagnostic limitations on patient outcomes.

Language: Minor grammatical and stylistic improvements are needed to enhance clarity and flow.

Response:

We thank the reviewer for this comments. We restructured the result and discussion sections and revised the manuscript accordingly as outlined above. We checked and corrected the language in the manuscript.

Overall Assessment:

Comment:

The paper is scientifically sound and makes a valuable contribution to the field. The required revisions are relatively minor and primarily focus on improving clarity, structure, and contextualization.

Response: We thank the reviewer for reviewing this manuscript

Reviewer #2: The study addresses an important gap in understanding critically ill patient outcomes in rural HDUs, making it highly relevant for clinicians and policymakers. Here are my recommendations:

Comment:

1. The introduction focuses heavily on HDU setup rather than framing a clear research gap.

Response:

We thank the reviewer for this comment: We clarified research gab and objectives in the introduction section (page 3 line 98-102.

Comment:

2. The objective should be explicitly stated earlier in the introduction and aligned more clearly with the research question.

Response:

We stated the objectives at the end of the introduction section more clearly on page 3 line 99-102.

Comment:

3. The methods section needs more focus on patient

---

## [Editor Report · Decision Letter 2]

Dear Dr. Rohacek,

Thank you for submitting your manuscript to PLOS ONE. After careful consideration, we feel that it has merit but does not fully meet PLOS ONE’s publication criteria as it currently stands. Therefore, we invite you to submit a revised version of the manuscript that addresses the points raised during the review process.

We look forward to receiving your revised manuscript.

Kind regards,

Abraham Aregay Desta, MSc.

Academic Editor

PLOS ONE

Journal Requirements:

Additional Editor Comments:

PONE-D-25-05201R2

Diagnoses and Critical Care Outcomes in a Rural Tanzanian High Dependency Unit: A Prospective Cohort Study

PLOS ONE

Dear Authors,

Thank you for submitting the 2nd round of the revised version of your manuscript entitled "Diagnoses and Critical Care Outcomes in a Rural Tanzanian High Dependency Unit: A Prospective Cohort Study" to PLOS One. We appreciate the time and effort you have invested in addressing the comments provided by the reviewers’ and editor during the previous rounds.

after carefully reviewing your updated manuscript and your detailed comments, your work is almost ready for publication. However, we kindly ask that you to address a few items before we move forward with acceptance. These are mostly clarifications or editorial adjustments that should be straightforward to address.

We kindly ask that you:

• Be transparent the scope of the generalizability as far as you included all the subjects in the sampling

• Make uniform editing throughout the manuscript such as line spacing and others

• Abstract should be a maximum of 300 words

• Revise the manuscript according to the PLOS One guideline

• Ensure to provide adequate response to all feedback given by the reviewers and editor

• Revise your manuscript considering the comments

• Provide a brief point-by-point response to each comment

• Highlight all changes made in the manuscript

Please submit your revised manuscript within 2 weeks. However, if you need additional time, do not hesitate to contact us.

We look forward to receiving your revised submission and moving forward with publication.

Regards,

Abraham Aregay Desta, MSc, PhD candidate.

PLOS ONE Editor

---

## [Author Response · Author response to Decision Letter 3]

25 Apr 2025

To April 25th 2025

Dr Abraham Aregay Desta

Editor

PLOS one

Dear Dr Desta

We thank you and the reviewers very much for the review of our manuscript

PONE-D-25-05201

"Diagnoses and Critical Care Outcomes in a rural Tanzanian High Dependency Unit: A Prospective Cohort Study"

We revised the manuscript according to the valuable comments of the reviewers and responded in a point-by-point reply below.

We uploaded a clean version of the manuscript and a version with track changes.

We uploaded the minimal dataset to https://zenodo.org/records/15177213

Martin Rohacek received funds from the Else Kröner Fresenius Foundation, Germany, Julie Rossier received funds from the Freiwillige Akademische Gesellschaft Basel, Switzerland. The funders had no role in study design, data collection and analysis, decision to publish, or preparation of the manuscript.

Thank you very much for evaluating our revised version of the manuscript.

Sincerely yours

PD Dr med Martin Rohacek

Swiss TPH

Kreuzstrasse 2

4123 Allschwil

Switzerland

University of Basel, Switzerland

Ifakara Health Institute, Tanzania

martin.rohacek@swisstph.ch, mrohacek@ihi.or.tz

Point by Point reply to editor’s comments in the e-mail from April 24th

Thank you for submitting the 2nd round of the revised version of your manuscript entitled "Diagnoses and Critical Care Outcomes in a Rural Tanzanian High Dependency Unit: A Prospective Cohort Study" to PLOS One. We appreciate the time and effort you have invested in addressing the comments provided by the reviewers’ and editor during the previous rounds.

after carefully reviewing your updated manuscript and your detailed comments, your work is almost ready for publication. However, we kindly ask that you to address a few items before we move forward with acceptance. These are mostly clarifications or editorial adjustments that should be straightforward to address.

We kindly ask that you:

Comment: Be transparent the scope of the generalizability as far as you included all the subjects in the sampling

Response: We thank the editor for this comment. While inclusion of all subjects eliminates sampling error, it does not guarantee generalizability to other populations. We revised line 431-432 on page 24 as follows: Last, this was a single centre study, and generalisability of these findings to populations living in other settings might be limited.

Comment: Make uniform editing throughout the manuscript such as line spacing and others

Response: We double spaced the whole manuscript and edited the font size according to PLOS one guidelines

Comment: Abstract should be a maximum of 300 words

Response: We shortened the Abstract to 299 words

Comment: Revise the manuscript according to the PLOS One guideline

Response: We revised the manuscript according to PLOS one guidelines

Comment Ensure to provide adequate response to all feedback given by the reviewers and editor

Response: We responded to all points raised in the point by point reply and revised the manuscript accordingly

Comment Revise your manuscript considering the comments

Response: The manuscript has been revised according to the comments of reviewers and editor

Comment: Provide a brief point-by-point response to each comment

Response: We provided a point by point response to each comment

Comment: Highlight all changes made in the manuscript

Response: We uploaded a tracked changed version

Comment: Please submit your revised manuscript within 2 weeks. However, if you need additional time, do not hesitate to contact us.

We look forward to receiving your revised submission and moving forward with publication.

Response: Thank you for reviewing our manuscript

Point By Point reply to reviewers’ comments which was attached to the e-mail from March 19th and April 1st

A Review for Manuscript Number PONE-D-25-05201

"Diagnoses and outcomes of critically ill patients admitted to a high dependency unit of a rural referral hospital in Tanzania: A prospective cohort study"

Title

Current Title: Diagnoses and outcomes of critically ill patients admitted to a high dependency unit of a rural referral hospital in Tanzania: A prospective cohort study

Comments:

Consider making the title more concise. For example:

"Critical Care Outcomes in a Rural Tanzanian High Dependency Unit: A Prospective Cohort Study"

Alternatively, emphasize the novelty of the study:

"First Insights into Critical Care Outcomes in a Rural Tanzanian High Dependency Unit: A Prospective Cohort Study"

Response:

We thank the reviewer for this comment. We changed the title to "First Insights into Critical Care Outcomes in a Rural Tanzanian High Dependency Unit: A Prospective Cohort Study" as suggested, but changed the title to "Diagnoses and Critical Care Outcomes in a rural Tanzanian High Dependency Unit: A Prospective Cohort Study" later.

Abstract

Comments:

Background: Briefly mention the gap in literature (e.g., lack of data on HDUs in rural sub-Saharan Africa).

Response:

We thank the reviewer for this comment. We revised the background of the abstract as follows:

Background: Data on rural sub-Saharan African high-dependency units (HDU) are lacking. We describe patient’s characteristics, diagnoses, and outcomes of patients admitted to a Tanzanian HDU, and identified factors associated with in-hospital mortality.

(page 2, line 41 – 43)

Methods:

Comment:

Clarify the sample size earlier (e.g., “491 patients admitted to the HDU”).

Response: We included this information into the first sentence of the methods section on page 4, line 95.

Results:

Comment:

Highlight the most striking finding (e.g., “Mortality during HDU stay was 30%, with sepsis and stroke being the deadliest conditions”).

Response:

We thank the reviewer for this comment. We moved the section “Patient outcomes and mortality” up to page 16 line 267 ff, adapted the numbers of S1-S3 Table and added the sentence “The most common deadliest conditions were sepsis, stroke, seizures, or aspiration pneumonia, with mortality rates of 51 to 65%, while mortality of patients with heart failure was 27% on page 16 line 269-271.

Conclusion:

Comment:

Add a sentence on implications for policy or practice (e.g., “These findings underscore the need for improved critical care resources in rural settings”).

Response:

We thank the reviewer for this suggestion, which we added to the conclusion on page 24 line 439-440.

Introduction

Comment:

Contextualize the Problem: Add a sentence on the burden of non-communicable diseases (NCDs) in Tanzania, as they are a major focus of the study.

Response:

We thank the reviewer for this comment.

We added information about NCDs to the first sentence of the introduction on page 3 line 66-69: Critical care medicine is important to manage seriously ill patients suffering from sepsis, pneumonia, and from non-communicable diseases (NCDs) such as heart failure and stroke. Globally, NCDs killed at least 43 million people in 2021, and 73% occurred in low- and middle-income countries.

Comment:

Justify the Study: Emphasize why this study is novel (e.g., “This is the first study to describe outcomes in a rural HDU in sub-Saharan Africa”).

Response:

We added the following sentence at the end of the Introduction: “This is the first study on outcomes of patients admitted to a HDU in rural sub-Saharan Africa”. On page 3 line 102-103.

Comment:

Clarify Objectives: Rephrase the objectives to be more specific (e.g., “To describe patient characteristics, diagnoses, and outcomes, and identify predictors of in-hospital mortality”).

Response:

We rephrased the objectives: “The objectives of this study were to describe characteristics, diagnoses, and outcomes of critically ill patients admitted to a recently implemented HDU of a referral hospital in rural Tanzania, and to describe predictors of in-hospital mortality” in the Introduction section on page 3 line 90-91.

Methods

Comment:

Study Design: Clarify the prospective nature of the study earlier in the section.

Response:

We thank the reviewer for this comment.

We included this into the first sentence in the methods section “This prospective observational single center cohort study including 491 patients was conducted at the HDU of the St. Francis Regional Referral Hospital (SFRRH), Ifakara, Tanzania” on page 4 line 95-96.

Comment:

Data Collection: Provide more detail on how data quality was ensured (e.g., training of data collectors, use of standardized forms).

Response:

We added the following sentence in the section study procedures and data collection on page 7 line 163-166:

“Data were collected by clinicians working at the HDU, and responded to queries raised by the data manager and the statistician who cleaned the data. Before the start of the study, all members were trained and instructed how to fill data into the standardized electronic data collection tools”

Comment:

Statistical Analysis: Briefly explain why specific statistical methods were chosen (e.g., Cox regression for mortality predictors).

Response:

We thank the reviewer for the suggestion, we have revised the Statistical Analysis section to clarify the rationale for the selected methods. The updated section now explicitly states why each statistical approach was chosen:

• Descriptive statistics were used to summarize baseline characteristics and diagnoses, as they provide a clear and concise overview of patient demographics and clinical profiles.

• Kaplan-Meier survival curves were used to visualize mortality rates over time, which is an effective method for estimating survival probabilities.

• Cox proportional hazards regression was utilized to identify factors associated with in-hospital mortality. This method was chosen because it accounts for varying follow-up times among patients and provides adjusted hazard ratios, which allow for meaningful comparisons of risk factors.

• AUROC analysis was performed to evaluate the predictive accuracy of existing mortality scores, as it is a well-established method for assessing the discriminative ability of clinical prediction models.

These statistical approaches were selected to ensure that our findings are both robust and clinically meaningful. We have incorporated these revisions into the manuscript.

We revised the statistical analysis section accordingly on page 8 line 195-215.

Comment:

Ethics: Mention how oral consent was documented to address potential concerns about consent validity.

Response:

We thank the reviewer for this comment.

Written informed consent was waived by both ethic committees. If patients were conscious, they were informed that data would be used for research purposes. In case of unconsciousness, relatives were informed. However, it was not specifically documented if the patient or the relative was informed (page 9 line 218-222).

Results

Comments:

Clarify Key Findings: Highlight the most important results in the text (e.g., “Sepsis and stroke were associated with the highest mortality rates”).

Response:

Thank you for this comment, the section about outcomes was moved up to page 16 line 267ff, and the sentence “The most common deadliest conditions were sepsis, stroke, seizures, or aspiration pneumonia, with mortality rates of 51 to 65%, while mortality of patients with heart failure was 27%” was added at page 16 line 269-271

Comment:

Simplify Tables: Consider merging or simplifying tables to improve readability (e.g., combine Tables 1 and 2).

Response:

We thank the reviewer for this comment. We simplified Table 1 and 2, however, we could not merge Table 1 and 2 together because table 1 shows baseline characteristics and Table 2 shows results

Comment:

Visuals: Ensure figures (e.g., ROC curves) are clearly labelled and interpretable.

Response:

We revised the labels and legends of the figures.

Discussion

Comment:

Structure: Use subheadings (e.g., “Key Findings”, “Comparison with Literature”, “Implications for Practice”) to improve flow.

Response: We added subheadings in the discussion section.

Comment:

Contextualize Findings: Discuss how the high mortality rates reflect systemic challenges in rural healthcare.

Response:

Thank you for this comment, we added this on page 23 line 389-394 “Late presentation and high mortality reflect systemic challenges in rural healthcare: The late presentation of patients in already serious conditions is due to lack of awareness of potentially life-threatening infectious – and non-communicable diseases in the communities, limited diagnostic and therapeutical options to diagnose serious conditions in the periphery, lack of transport, and the fact that a majority of patients do not have a health insurance to cover the costs”.

Comment:

Policy Implications: Add a paragraph on how the findings can inform policy (e.g., “These results highlight the need for increased investment in rural critical care infrastructure”).

Response:

Thank you for this comment, we added more information on page 23 line 394-405.

Comment:

Limitations: Expand on the impact of lost-to-follow-up cases and diagnostic limitations.

Response:

Thank you for this comment, we expanded this in the limitation section, page 24 line 425-427:

“Third, 9% of patients were lost to follow-up after discharge from the hospital, and patients could not be followed for a longer period to determine their post discharge outcome. However, we could analyse all patients for the endpoint of in-hospital death”.

Conclusion

Comment:

Call to Action: Add a sentence on next steps (e.g., “Future studies should explore interventions to reduce mortality in rural HDUs”).

Response: Thank you for this comment, We added this sentence on page 25 line 441.

Comment:

Broader Impact: Mention how the study contributes to global health equity (e.g., “This study provides critical insights into improving healthcare access in low-resource settings”).

Response:

We added “These findings underscore the need for improved critical care in low-resource rural settings on page 18 line 439-440.

References

Comment:

Ensure all references are formatted consistently according to the journal’s guidelines.

Response:

We checked all references and formatted accordingly

Comment:

Include more references from sub-Saharan Africa to strengthen the regional context.

Response:

We added the following References in the discussion section

Beaney T, Burrell LM, Castillo RR, et al. May Measurement Month 2018: a pragmatic global screening campaign to raise awareness of blood pressure by the International Society of Hypertension. Eur Heart J 2019;40:2006-2017. doi: 10.1093/eurheartj/ehz300

WHO PEN and integrated outpatient care for severe, chronic NCDS at first referral hospitals in the African region (PEN-PLUS). 2019.

https://www.afro.who.int/publications/who-pen-and-integrated-outpatient-care-severe-chronic-ncds-first-referral-hospitals. Last access March 24th 2025

Klassen SL, Okello E, Ferrer JME, et al. Decentralization and Integration of Advanced Cardiac Care for the World's Poorest Billion Through the PEN-Plus Strategy for Severe Chronic Non-Communicable Disease. Glob Heart 2024;19:33. doi: 10.5334/gh.1313

WHO. HEARTS; Technical package for cardiovascular disease management in primary health care: Tool for the development of a consensus protocol for treatment of hypertension: technical package for cardiovascular disease management in primary health care; https://www.who.int/publications/i/item/WHO-NMH-NVI-19-8. Last access March 24th 2025

Kivuyo S, Birungi J, Okebe J, et al. Integrated management of HIV, diabetes, and hypertension in sub-Saharan Africa (INTE-AFRICA): a pragmatic cluster-randomised, controlled trial. Lancet 2023;402:1241-1250. doi: 10.1016/S0140-6736(23)01573-8

Baker T, Scribante J, Elhadi M, Ademuyiwa A, Osinaike B, Owoo C, et al. The African Critical Illness Outcomes Study (ACIOS): a point prevalence study of critical illness in 22 nations in Africa. The Lancet. 2025;405: 715–724. doi:10.1016/S0140-6736(24)02846-0

Overall Suggestions

Language and Clarity:

Comment:

• Simplify complex sentences for better readability.

• Avoid jargon and define acronyms (e.g., HDU,

---

## [Editor Report · Decision Letter 3]

Diagnoses and Critical Care Outcomes in a Rural Tanzanian High Dependency Unit: A Prospective Cohort Study

PONE-D-25-05201R3

Dear Authors,

We’re pleased to inform you that your manuscript has been judged scientifically suitable for publication and will be formally accepted for publication once it meets all outstanding technical requirements.

Kind regards,

Abraham Aregay Desta

Academic Editor

PLOS ONE

---

## [Editor Report · Acceptance letter]

PONE-D-25-05201R3

PLOS ONE

Dear Dr. Rohacek,

I'm pleased to inform you that your manuscript has been deemed suitable for publication in PLOS ONE. Congratulations! Your manuscript is now being handed over to our production team.

Kind regards,

on behalf of

Dr. Abraham Aregay Desta

Academic Editor

PLOS ONE